# Leveraging Instruction Language Model to Generate Vectorized RISC-V Tensor Programs

## Abstract

Auto-vectorization is a powerful optimization that significantly improves the performance of tensor programs on modern instruction set architectures. Transforming tensor programs into high performance vectorized programs is in high demand. However, traditional compilers often overlook opportunities to vectorization. Meanwhile, hand-crafting vectorized optimization using specialized instructions remains a complicated and error-prone endeavor that requires in-depth knowledge of specific instruction set architectures and compilers. In this paper, we introduce *RISCompiler*, a compiler designed to generate vectorized tensor programs with auto-vectorization tailored for the RISC-V target with vector extension. The main concept involves transforming the tensor program exploration task into generation task exploiting an instruction language model (ILM). To facilitate this, we create an instruction sentence representation suitable for ILM, which includes transformation details to accurately represent vectorized RISC-V tensor programs. RISCompiler uses an innovative, parameter-efficient fine-tuning mechanism to enhance domain adaptation by strategically concentrating on vectorized components, thereby boosting both fine-tuning and inference efficiency. During the compilation, the ILM incorporates insights from offline learning and prior transformations to make optimal optimizations within the current design space. Experimental results demonstrate that RISCompiler, which are capable of generating high-performance vectorized programs automatically, surpasses existing state-of-the-art compilers and scalar versions by a substantial margin.

## 1 Introduction

In recent years, the enhancement of parallel computing capabilities has emerged as a crucial trend in improving the computational efficiency of high-performance computing (HPC) and artificial intelligence (AI) applications. This trend is particularly prominent in the realm of emerging instruction set architecture (ISA), where the open-source RISC-V ISA has been garnering increasing attention. RISC-V introduces a flexible vector extension (RVV) that supports single instruction, multiple data (SIMD) parallel computing. Notably, the vectorized instructions in RISC-V have been shown to significantly boost the performance of both floating-point and integer computations. By enabling operations to be executed on multiple cores or processing elements concurrently, vectorized instructions significantly enhance processing speed compared to unvectorized ones which handle operations sequentially and iteratively.

To enable programmers to directly exploit vectorization features, the RISC-V vector (RVV) extension introduces a collection of compiler intrinsics that enable the invocation of SIMD instructions at the assembly level directly from C source code. For instance, the instruction `vle32_v_f32m1` allows for loading a vector of 32-bit floating-point elements from memory into a vector register. The instruction `vfmacc_vv_f32m1` performs an element-wise multiplication of two input vector registers and accumulates the results into a third vector register. While `vse32_v_f32m1` enables to store a vector of 32-bit floating-point elements from a vector register back to memory. Given the complex semantics involved in these operations, the task of writing such programs is complicated, error-prone, and primarily the domain of experienced programmers. Therefore, significant research efforts have been devoted to automatic vectorization over several decades.

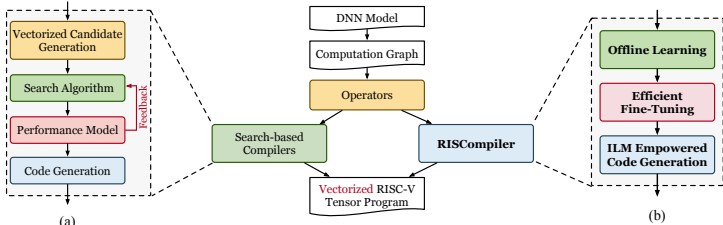

Figure 1: (a) The workflow of search-based tensor compilers with a learned performance model for vectorized programs. (b) Our proposed generation-based tensor compiler with three different stages for the automatic optimization of vectorized RISC-V programs.

However, for search-based tensor compilers, which are the majority of the present, their auto-vectorization performance is often hampered by the inherent inaccuracy of static analysis. One of major reasons lies in that the vectorization process relies on data dependency analysis, a task which is difficult to perform accurately due to the intricate nature of the code. All the factors, such as complex control flow patterns, aliasing issues, and discontinuous memory access patterns, need to be considered in the optimization process. Moreover, learned performance models, which are widely adopted by compilers to determine the outcomes of vectorization, often leads to suboptimal optimization decisions due to their limited versatility.

**Our Goal** Recent advances in LLMs have demonstrated the potential to generate and transform code based on natural language instructions Chen et al. (2021); Roziere et al. (2020). Motivated by these progress, we ask the following question to democratize the generation of vectorized RISC-V programs by non-experts: *Can we build a dedicated instruction language model to automatically optimize scalar RISC-V tensor programs into semantic equivalent vectorized versions?*

**Challenges** To design our generation-based RISC-V auto vectorization compiler utilizing instruction language model (ILM), we summarize the challenging aspects introduced by new design paradigm as follows: *Sufficient Transformation Space.* Instead of building a new transformation space for the RISC-V instruction specializations with search-based compilers, we need to create a language-model-friendly instruction representation that record transformation to represent vectorized programs, bridging the gap between specialized instructions and language models. *Vectorized Instruction Generation.* Given instruction sentences, the ILM should be able to determine the most suitable transformation for the RISC-V target, with the aim of producing valid vectorized programs within the transformation space. *Efficient Fine-Tuning for Language Model.* We need to propose an optimal fine-tuning strategy that can yield enhanced ILM performance while achieving efficiency with a minimized trainable parameter size.

**Our Solution** We design and implement *RISCompiler*, an end-to-end compilation framework to generate RISC-V tensor programs with auto-vectorization, overcoming three challenges mentioned above. We evaluate RISCompiler on standard deep learning benchmarks, including both single operator and end-to-end network against state-of-the-art compilers. Experiment results show that RISCompiler improves the execution performance of workloads across single operator to end-to-end network by up to $1.65\times$ and $1.38\times$, respectively. The contribution can be summarized as follows: (1) We exploit an instruction language model to assist auto-generating high-performance vectorized RISC-V programs, transforming the program exploration task into generation task. (2) We construct instruction sentences for language model to understand instruction information and vectorized transformation. (3) We design a novel fine-tuning algorithm that features an asymmetric structure with a shared part for all tensor programs and distinct parts for each vectorized instruction.

## 2 BACKGROUND AND RELATED WORK

**Search-based Tensor Compilers** Figure 1(a) shows the general workflow of a search-based compiler with a learned performance model. This paradigm is used by plenty of recent tensor compilers such as TVM, Halide and XLA Chen et al. (2018); Ragan-Kelley et al. (2012); XLA Team (2017). The compiler adopts a high-level mathematical expression to represent the operator as input and employs a search algorithm Baghdadi et al. (2021); Zheng et al. (2020a); Adams et al. (2019) to

find the optimal tensor program. The primary search spaces formed by derivation rules include loop optimizations (*e.g.*, tiling, parallelization, unrolling, fusion), which typically target general-purpose processor ISAs. During the search, the compiler generates promising candidates and evaluates their performance—either via a performance model or hardware execution. Given the large search space and time-consuming on-device measurements, a learned performance model is often used to guide the search, whose effectiveness critically determines the search's efficiency and quality.

**Auto-vectorization with Compilation** The RISC-V vector (RVV) extension is a set of architectural extensions to the basic RISC-V ISA that provides support for vector processing. Vector processing is a technique that enables the simultaneous execution of the same operation on multiple data elements, which can significantly improve the performance of the workloads. RVV defines a set of vector registers and a set of vector instructions that can operate on these registers. Meanwhile, the RVV is designed to be highly flexible and configurable, allowing programmers to choose the vector length and the supported data types based on the specific requirements of their application domain. TVM Chen et al. (2018), a search-based compiler, provides an extensible interface for SIMD instruction support. However, it requires programmers to manually adapt programs to match the behavior of target instructions and predefine lowering rules for specializations prior to compilation. To generate optimized LLVM IR Lattner & Adve (2004), TVM aligns computations and replaces program segments with ISA-specific code snippets. This process demands significant engineering effort, involving modifications across multiple layers of the intermediate representation (IR), as well as iterative passes and transformations. Finally, code generation is delegated to LLVM's backend.

**Parameter-efficient Fine-tuning** LLMs are extremely powerful in language processing tasks, yet their compatibility to one specific task heavily relies on fine-tuning technique, which typically demands substantial computation resources. This need catalyzes the investigation of parameter-efficient fine-tuning (PEFT) techniques, which aim to reduce the computational and storage requirements during model adaptation. Among the prominent PEFT methodologies are adapters Houlsby et al. (2019); Rebuffi et al. (2017), which incorporate new trainable dense layers within the extant model architecture, while preserving the original parameters in a frozen state. The adapter paradigm has demonstrated efficacy across a multitude of domains Pfeiffer et al. (2020); Stickland & Murray (2019); Sung et al. (2022); Zhou et al. (2024). Enhancements in adapter compactness are pursued through the construction of parameter matrices via the Kronecker product of low-rank matrices Karimi Mahabadi et al. (2021). An alternative PEFT strategy involves the direct manipulation of activations utilizing learned vectors, achievable via concatenation Liu et al. (2024); Li & Liang (2021); Lester et al. (2021), multiplication $IA^3$ Liu et al. (2022), or addition BitFit Zaken et al. (2021). Notable examples such as prefix-tuning Li & Liang (2021) and prompt-tuning Lester et al. (2021) fine-tune continuous prompts as opposed to crafting discrete ones. Intriguingly, numerous PEFT methods can be conceptualized as variants of adapters, offering a cohesive framework He et al. (2021). Beyond the mere augmentation of parameters or the alteration of the computation graph, scholarly efforts are also directed towards sparse Guo et al. (2020); Sung et al. (2021) or low-rank updates such as LoRA Hu et al. (2021).

# 3 DESIGN OVERVIEW OF RISCOMPILER

RISCompiler is an end-to-end compilation flow that leverages language models to perform auto-vectorization for RISC-V programs with vector extension. The compilation process begins with a prompt from programmers in natural language, which contains vectorized optimization request for a RISC-V hardware specification. Besides, a fundamental scalar tensor program snippet is also provided as the transformation target. Upon receiving these inputs, RISCompiler processes the request and subsequently generates the corresponding RVV instructions, utilizing ILM to generate complete RISC-V programs with auto-vectorization. Figure 1(b) and Figure 2 show the workflow of the RISCompiler which has three important stages: *i)* offline learning stage that constructs a transformation space and samples diverse vectorized RISC-V programs from it; *ii)* efficient fine-tuning stage that fine-tunes the ILM with the performance of sampled vectorized RISC-V programs and an efficient learning strategy; *iii)* code generation stage that generates high-performance RISC-V programs for the input operators with auto-vectorization.

**Offline Learning Stage** One of the primary challenges that RISCompiler has to address is making a comprehensive dataset for the fine-tuning of the ILM. We build an expansive transformation space

Figure 2: Three important stages in RISCompiler. During the offline learning stage, the language model-friendly instruction sentences that represent vectorized RISC-V programs are designed for the offline learning. During the efficient fine-tuning stage, an asymmetric LoRA structure first adaptively identifies and initializes multiple vectorized optimization components. Subsequently, it utilizes a trainable MoE router that considers each vectorized component as an expert to autonomously partition training samples for the ILM fine-tuning. During the code generation stage, ILM dynamically and flexibly merges multiple expert adapters through a trained router. Finally, the vectorized RISC-V programs are generated.

and design the language-model friendly instruction sentences that record transformations to represent vectorized RISC-V programs. During the compilation, ILM combines knowledge from offline learning and previously made transformations to sample the best decision in the current space.

**Efficient Fine-tuning Stage** Because the performance of ILM with only pre-trained weights may not be able to stay perfect all the time, we fine-tune ILM efficiently with the online dataset in the second stage. An asymmetric LoRA architecture is introduced to perform PEFT iteratively. Parts of ILM are selectively activated by a gating mechanism in response to different operator configurations with instruction sentences at each iteration. Meanwhile, ILM handles each vectorized instruction-dependent components as expert adapters to ensure computational efficiency throughout the fine-tuning.

**Code Generation Stage** During the code generation, ILM merges expert adapters by enabling routing computation based on the input operator and instruction sentences. It flexibly and dynamically merges multiple vectorized expert adapters through the router with Mixture-of-Experts (MoE) manner Shazeer et al. (2017).

## 4 OFFLINE LEARNING STAGE

**Vectorized Transformation Space** Figure 2 shows the three important stages of RISCompiler. The offline learning stage has three steps. The goal of step ① is to create a large space for RISC-V programs with vectorized optimizations to ensure high-performance. This space encompasses a range of potential optimizations, both unvectorized and vectorized, which entail tasks such as determining tiling sizes for loop axes, specifying unroll steps, choosing computation locations for operators, planning parallelization strategies, defining vector width, determining the number of registers utilized, and so forth. Therefore, the transformation space for the input operator can be formulated as follows:

$$R = \left\{ r^{(n)} \mid \begin{array}{l} r^{(i)} = \text{Transform}\left(r^{(i-1)}, t_i\right), \\ \forall t_i \in T_i, 1 \le i \le n \end{array} \right\}, \tag{1}$$

where $r^{(0)}$ indicates the scalar programs of the input operator and $t_i$ denotes a random optimization from the space $T_i$. The number of transformation combinations can be defined as:

$$|R| = |T_1| \times |T_2| \times \ldots \times |T_n|.$$

The options for each optimization create an optimization space, and all optimization combinations form the transformation space. The primary objective of the transformation space is to find an optimization combination that improves the execution of the programs.

We draw inspiration from the previous search-based tensor compiler Zheng et al. (2020a;b); Shao et al. (2022); Bai et al. (2023); Zhai et al. (2024) to construct the vectorized transformation space.

This is necessary because the existing space for scalar programs, particularly those not utilizing the RVV specifications, is already sufficiently large. Expanding upon the prior transformation space, we introduce specific optimizations tailored for RVV. These optimizations primarily encompass vector width, the count of registers, vector element count, vector instruction combinations, vector memory access patterns, vector scheduling, and vector precision.

**Instruction Sentences** The objective of step ② is to delineate vectorized RISC-V programs selected from the transformation space. As elucidated in Section 4, vectorized RISC-V programs $r^{(n)}$ comprises the initial program $r^{(0)}$ of the operator and transformations $t_1, t_2, \ldots, t_n$. Each transformation $t_i$ is stochastically chosen from the space $T_i$, focusing on optimizations related to RVV. Directly generating RISC-V assembly programs with language models inherently poses chal-

```
1  p0 p1 matmul p2 00a625a8 1024 512 1024 512 1024 512
2  riscv -keys=cpu -mcpu=rv64gcv -model=rv64 SP 2 0 1024 32 1 4 1
3  SP 2 4 512 8 1 4 1 SP 2 8 1024 1024 1 RE 2 0 4 1 5 8 2 6 9 3 7
4  FSP 4 0 0 2 FSP 4 3 1 2 RE 4 0 3 1 4 2 5 CA 2 4 3
5  vector_width$8,16,32,64
6  vector_reg_count$16,32,64
7  Vector_element_count$4,8,16
8  vector_instr_combo$vadd+vmul,vle+vse,...
9  vector_mem_access$continuous,noncontinuous
10 vector_sched$static,dynamic
11 vector_precision$int8,int16,float32
12 auto_unroll_max_step$0 ESC
```

Figure 3: The sample of instruction sentence designed for ILM including input operator and vectorized optimizations.

lenges. The length of RISC-V assembly programs typically exceeds ten thousand tokens, and strict adherence to syntactic rules is required. Crafting such extensive and syntactically correct RISC-V assembly programs is nearly unfeasible. Hence, rather than striving for the direct end-to-end generation of assembly programs, we take advantage of the scalar version of tensor programs that represent the input operator and language models to assist in transformation-making. To facilitate this, we design instruction sentences which is amenable to language models, enabling the recording of transformation to represent vectorized RISC-V programs.

Figure 3 showcases a sample of instruction sentence designed for vectorized RISC-V programs. The red portion signifies the names of operators and the dimensions of their corresponding inputs $p_{0,1}$ and outputs $p_2$, while the black section pertains to instruction specializations tailored for RISC-V. The green and blue segments correspond to the transformations for the vectorized optimization. For instance, the dynamic adjustment of the vector width is achieved using the `vsetvl` instruction to align with the computational requirements of each operation, efficiently utilizing the available vector registers. The optimization of register count is also paramount, meticulously managing the utilization of the ample vector registers. Tuning the vector element count is essential to strike a delicate balance between computational intensity and memory bandwidth constraints. Additionally, the selection of vector instruction combinations, such as fused multiply-add (`vadd+vmul`, `vfmacc`) and other specialized vector instructions, is crucial in optimizing performance by reducing the number of required operations.

**Offline Dataset** In step ③, we conduct an extensive random sampling of the transformation space in order to acquire an offline dataset from which the ILM can derive knowledge. This process serves the following objectives: Firstly, ensuring that the samples from the dataset align with the distribution in the transformation space; Secondly, expanding the vocabulary comprehensively to encompass all vectorized transformation possibilities, such as "vector width" or "vector register count" which are tokenized into distinct tokens. The operators sampled for ILM draw inspiration from TenSet Zheng et al. (2021). These operators are sourced from PyTorch Image Models (timm) Wightman (2019) and Huggingface's Transformer Models Wolf et al. (2020), covering tasks in both computer vision and natural language processing. We modify the input shape to create a diverse range of operators with small batch sizes for constructing the offline dataset. The dataset comprises 86 workloads and around 1.8K operators, with 0.5 million instruction sentences encompassing vectorized optimizations. It takes about 8.5 hours to collect dataset with a 16-core server.

## 5 INSTRUCTION GENERATION

Taking into consideration both learning capabilities and resource constraints, ILM adopts the model architecture of Llama-3.2-3B IIa. We fine-tune ILM in a supervised manner with the offline dataset.

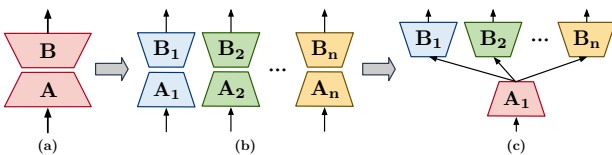

Figure 4: (a) The LoRA structure encompasses a low rank matrix $\vec{A}$ and $\vec{B}$. (b) LoRA is partitioned into numerous smaller $\vec{A}$ and $\vec{B}$ matrices with an equivalent parameter count to avoid training interference. (c) Our design exhibits an asymmetric structure featuring a shared $\vec{A}$ matrix and multiple $\vec{B}$ matrices.

## 5.1 EFFICIENT FINE-TUNING STAGE

LoRA Hu et al. (2021) attains performance levels comparable to fine-tuning across numerous benchmarks by maintaining the pre-trained language model $\vec{W}_0$ in a static state and integrating trainable low-rank decomposition matrices, $\vec{A}$ and $\vec{B}$, within each layer. The forward computation is expressed as follows:

$$\vec{y}' = \vec{y} + \Delta\vec{y} = \vec{W}_0\vec{x} + B\vec{A}x, \tag{2}$$

where $y \in \mathbb{R}^d$ represents the output and $x \in \mathbb{R}^k$ denotes the input. The matrices $\vec{B} \in \mathbb{R}^{d \times r}$ and $\vec{A} \in \mathbb{R}^{r \times k}$ are defined with $r \ll \min(d, k)$. Typically, matrix $\vec{B}$ is initialized with zeroes, while matrix $\vec{A}$ via the Kaiming Uniform He et al. (2015) to ensure that $\Delta\vec{y} = 0$ at the outset.

As for ILM, we aim for a PEFT approach that strikes a better balance between maximizing the learning capability for vectorized programs and minimizing the computation. A promising approach involves partitioning LoRA into multi-structured variants, characterized by a central and shared matrix $\vec{A}$ coupled with multiple distinct matrices $\vec{B}$, which facilitates a synergy of shared knowledge for the tensor programs and vectorized transformation. The asymmetric LoRA structure shown in Figure 4(c) can be formulated as:

$$\vec{W} = \vec{W}_0 + \Delta\vec{W} = \vec{W}_0 + \sum_{i=1}^{N} \omega_i \cdot \vec{B}_i\vec{A}. \tag{3}$$

The matrices $\vec{B}_i \in \mathbb{R}^{d \times r}$ and shared $\vec{A} \in \mathbb{R}^{r \times k}$. The hyperparameter $N$ denotes the number of $\vec{B}$ matrices. The term $\omega_i$ modulates these contribution weights for head $\vec{B}_i$. In order to achieve a unified approach to the distinct forward processes of multiple $\vec{B}$ matrices, we define a set of exports denoted as $(\vec{E}_1, \vec{E}_2, \ldots, \vec{E}_N)$ to learn the updated matrix $\Delta\vec{W}$ for the vectorized transformation. Based on this design, the forward process of our proposed method is expressed as:

$$y = \vec{W}_0\vec{x} + \sum_{i=1}^{N} \omega_i\vec{E}_i\vec{A}\vec{x}, \qquad \text{(MoE)} \tag{4}$$

where $N$ denotes the number of experts. We introduce a gate function commonly consisting of a dense layer with trainable weights $\vec{W}_g \in \mathbb{R}^{r \times N}$ followed by a softmax function which takes an intermediate token representation $\vec{x}$ as input and combines the output of each expert based on the gating scores $(\omega_1, \ldots, \omega_N)$:

$$\omega_i = \text{softmax}(\vec{W}_g^{\top}\vec{x}). \qquad \text{(Router)} \tag{5}$$

During the fine-tuning, the weights of ILM remain frozen, while the experts and router layers are trained from scratch.

## 5.2 CODE GENERATION STAGE

The RISCompiler utilizes the ILM for transformation-making in the generation of vectorized RISC-V programs. Upon applying a transformation, the ILM assesses its compatibility within the space.

Should the transformation be deemed invalid, the ILM discards the programs and commences the regeneration. Given that the transformations are acquired via sampling, the subsequent generation may explore alternative decisions, thereby averting persistent failures in the regeneration.

Figure 5 illustrates the process of vectorized optimization along a specific axis for a matrix multiplication on the RISC-V target. Initially, the instruction sentence encompasses the input operator, vectorized RISC-V instruction specifications, and the representation of $i = 1024$, indicating the length of a specific axis within the operator. These components are consolidated to serve as the input prompt for the ILM. The ILM utilizes the prompt information to probabilistically generate the next token. For instance, in the context of "vector width" optimization, it may generate "$i.0 = 32$", indicating the instruction to set the vector width to 32 bits.

The subsequent step involves combining the prompt with the token predicted in the previous step regarding vector width. This combined information forms a new input prompt that is fed into the ILM to make predictions for the next token. Successively optimizing different aspects of vectorized RISC-V optimizations, such as the number of "vector registers" with "$i.1 = 1$", and the "vector element count" for parallel processing with "$i.2 = 4$", is carried out to optimize the specific axis of the operator. This iterative process continues, enabling the comprehensive optimization of the entire operator.

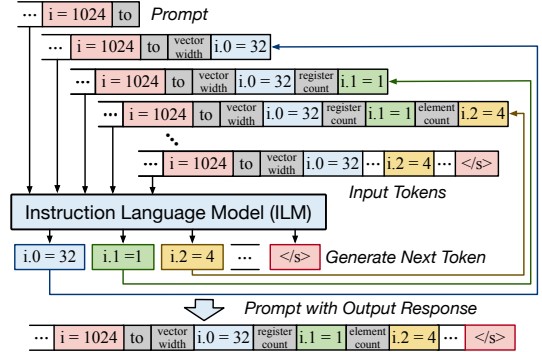

Figure 5: Illustration of generating instruction sentences with vectorized optimizations.

It is worth noting that the ILM does not directly take the entire operator and specific vectorized RISC-V specifications as input to generate all transformation in one go. Instead, to enhance the stability and usability of the ILM, we focus solely on optimizing one subset of transformations along the specific axis of the operator at a time which efficiently sieves out invalid optimization.

## 6 EVALUATION

Experiments are conducted on a server equipped with a 16-core, 24-thread Intel i9-12900K CPU (hyper-threading) along with two 4090D GPUs (48GB), operating on Ubuntu LTS 22.04. Fine-tuning the ILM for 4 epochs typically requires approximately 35 hours to complete. RISC-V toolchain is constructed via LLVM Clang llv (b). Our experiments are con-

Table 1: GEMV/GEMM Configuration.

| Category | M | N | K | #Test Cases |
|---|---|---|---|---|
| Operators | [1, 256] | [1, 256] | [1, 256] | 50 |
| | [1, 256] | [1, 256] | [257, 65536] | 44 |
| | [257, 1024] | [1, 256] | [1, 65536] | 32 |
| | [257, 1024] | [257, 65536] | [1, 65536] | 59 |
| | [1025, 8192] | [1, 256] | [1, 65536] | 33 |
| | [1025, 8192] | [257, 8192] | [1, 65536] | 64 |

ducted using a specific version of LLVM as indicated by commit llv (a), and our evaluation pertains to the auto-vectorization for RVV version 1.0. To conduct a performance assessment of RISCompiler in comparison to the state-of-art compilers, we employ the LLVM Clang (LLVM-Vectorizer), which is integrated into the GNU toolchain comprising libgcc, the GNU linker, and C libraries. Additionally, we utilize GCC-Loops version 12.2.0. The results obtained from GCC-Loops are compiled using Clang and then linked using riscv64-unknown-elf-ld. All experimental results are validated using the QEMU emulator Bellard (2005) with the RISC-V Vector Extensions. The baseline methodology incorporates the Claude-3-Haiku with Poe poe without any fine-tuning technique as its central component to generate programs with vectorized programs. We use floating point 32 as the data type and set the number of experts at 8 during efficient fine-tuning stage.

### 6.1 SINGLE OPERATOR BENCHMARK

**Workloads** We begin by evaluating RISCompiler on a range of common deep learning operators, using a suite of standard machine learning workloads, encompassing one-dimensional (Conv-1D),

Table 3: Comparative performance of fine-tuning strategies.

| Perf. | LoRA Hu et al. (2021) | AdaLoRA Zhang et al. (2023) | LoRA-Hub Huang et al. | LoRA-MoE Dou et al. (2023) | Ours |
|---|---|---|---|---|---|
| **Top-1** | 0.7233 | 0.7564 | 0.7975 | 0.8412 | 0.8857 |
| **Top-5** | 0.9134 | 0.9228 | 0.9255 | 0.9261 | 0.9416 |

two-dimensional (Conv-2D), depth-wise (DEP) convolution operators, general matrix multiplication (GEMM), and general matrix-vector multiplication (GEMV). Table 1 and Table 2 present the benchmarks for these operators. Each test case is characterized by a unique shape size, and the range of values is denoted as $[min, max]$. For matrix operations, we consider a total of 733 test cases derived from real-world applications, specifically Transformer-based models such as BERT Devlin et al. (2018), DistilBERT Sanh et al. (2019), RoBERTa Liu et al. (2019), Llama2-7B Touvron et al. (2023), GPT-2 and ViT-B/16 Dosovitskiy et al. (2021). For each operator, a selection of specific shape configurations is undertaken for analysis, and these configurations are evaluated across three distinct batch sizes: 1, 8, and 16.

**Performance Results** As shown in Figure 6, RISCompiler performs the best in all single operators and batch size settings. RISCompiler outperforms existing state-of-the-art compilers with auto-vectorization optimizations such as GCC-Loops and LLVM-Vectorizer by up to $1.65\times$ and $1.24\times$. The performance improvements of RISCompiler come from both its instruction language model and effective parameter-efficient fine-tuning strategy. For most operators, we find that the best vectorized

Table 2: Convolution/Depthwise Configuration.

| Category | Fmap Size | Filter Size | #Test Cases |
|---|---|---|---|
| **U-Net** | [280, 568] | 3×3 | 40 |
| | [56, 138] | 3×3 | 44 |
| | [8, 52] | 1×1 | 32 |
| **ResNet-50** | [56, 120] | 1×1 | 40 |
| | [32, 80] | 3×3 | 60 |
| | [4, 40] | 1×1 | 34 |
| | [2, 30] | 3×3 | 70 |
| **RetinaNet** | [2, 28] | 1×1 / 3×3 | 49/59 |
| **MobileNet-V2** | [14, 112] | 3×3 | 23 |

tensor programs generated by RISCompiler is outside the decision space of existing language model Claude-3-Haiku because the ILM of RISCompiler is able to explore more vectorized optimization combinations with efficient fine-tuning algorithm. For instance, Claude-3-Haiku can only achieve a $1.34\times$ improvement compared to the standard scalar versions within these operators while the ILM of RISCompiler can achieve a $2.99\times$ enhancement.

**Ablation Study** We run four variants on Conv-2D operators and report the performance. We pick the last convolution operator in ResNet-50 with batch size 8 as the test case. In Table 4, each setting corresponds to different variants. Setting#(d) uses all our introduced methods. Setting#(a) means the direct utilization of the standard scalar version of the RISC-V programs implemented with corresponding operators, without involving any vectorized optimizations or the intervention of language models. Setting#(b) represents solely utilizing an offline dataset to train a LlaMA-3.2-1B language lla model with the standard LoRA fine-tuning in offline learning stage, specifically for optimizing the RISC-V Vector Extension. The primary distinction between Setting#(c) and Setting#(b) lies in the utilization of the LlaMA-3.2-3B as the instruction language model. Overall, when combining our proposed efficient fine-tuning algorithm with the corresponding instruction language model through offline dataset, good final performance can be achieved.

## 6.2 END-TO-END NETWORK BENCHMARK

**Workloads** We conduct a comprehensive benchmark of the end-to-end performance of various DNNs with batch size 1, which encompass ResNet-50, VGG-16, MobileNet-v2 Sandler et al. (2018), DenseNet Huang et al. (2017) for image classification, as well as BERT-based and Bert-Tiny Devlin et al. (2018) for language understanding.

**Quality** As for the fine-tuning quality, we hold out a test set that consists of four DNNs: U-Net, ResNet-50, MobileNet-V2 and Bert-Base with batch size 1. Divide all the remaining data into training and validation sets at a ratio of 8:2. We use the Top-k score as evaluation metric which can be represented as: $top - k = \frac{\sum_m \sum_s min\_cycle_{m,s} \times weight_{m,s}}{\sum_m \sum_s \min(cycle_{m,s,i}) \times weight_{m,s}}, 1 \leq i \leq k$. Specifically, $min_cycle$ represents the minimum simulated cycle for operator $s$ in DNN $m$, $weight_{m,s}$ denotes the frequency of occurrences of operator $s$ in DNN $m$, and $cycle_{m,s,i}$ indicates the simulated cycles

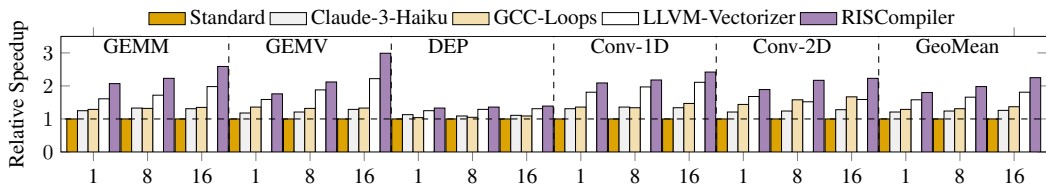

Figure 6: Speedup of single operator performance normalized by the standard scalar version.

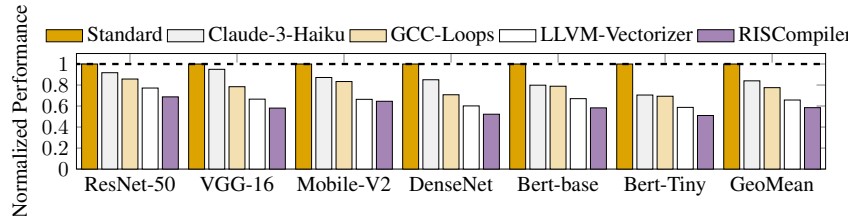

Figure 7: End-to-end performance comparison of different baselines.

corresponding to the $i-$th largest value of the output score generated by the ILM for operator $s$ in DNN $m$. Our proposed asymmetric LoRA structure outperforms the baseline LoRA fine-tuning (LoRA), Lora-Hub learning, multi-LoRA tuning with MoE inference, and AdaLoRA algorithms based on the Top-1 and Top-5 scores as evaluation metrics. Further details are available in Table 3.

**Performance Results** Figure 7 shows the normalized performance. For each end-to-end test case, we normalize the performance to the standard scalar version of the tensor programs. RISCompiler outperforms the best in all test cases and outperforms existing state-of-the-art compilers such as GCC-Loops and LLVM-Vectorizer by up to $1.36\times$ and $1.15\times$, respectively. In comparison to the common Claude-3-Haiku language model, RISCompiler can achieve a maximum acceleration of up to $1.38\times$ on these DNN benchmarks.

### 6.3 DISCUSSION

One limitation of RISCompiler is the necessity to employ various performance-oriented and vectorized programs during the fine-tuning stage. Therefore, the sampling of high-quality programs is particularly crucial. Another limitation is that RISCompiler prioritizes efficiency during the pre-

Table 4: Ablation study on a convolution operator.

| Setting | RISCompiler | | | |
|---|---|---|---|---|
| | (a) | (b) | (c) | (d) |
| **Offline Dataset** | | ✓ | ✓ | ✓ |
| **Instruction Language Model** | | | ✓ | ✓ |
| **Efficient Fine-Tuning** | | | | ✓ |
| **Perf.** | $1.00\times$ | $1.71\times$ | $2.04\times$ | $2.36\times$ |

compilation phase. The substantial time commitment required for fine-tuning the ILM, which could span over numerous hours, should not be underestimated. If the goal is limited to compiling a single model or a small set of operators, the ILM may not be the most cost-effective choice. However, it is more suitable for compiling a diverse range of models with vectorized optimizations.

## 7 CONCLUSION

We present RISCompiler, an innovative compilation flow that incorporates an instruction language model with a parameter-efficient fine-tuning strategy for generating vectorized RISC-V programs. RISCompiler is notable for its pioneering integration of language models into the compilation domain, providing auto-vectorization support for the RISC-V target. By harnessing the combined strengths of offline dataset learning and efficient fine-tuning, RISCompiler demonstrates robust capabilities in generating vectorized RISC-V tensor programs on modern deep learning benchmarks.

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
