## A   IMPLEMENTATION DETAILS

### A.1   INSTRUCTION SENTENCES

As detailed in Section 4, RISCompiler constructs Instruction Sentences through a structured process illustrated in Figure 8 (identical to Figure 3 in the main text). These sentences serve as intermediate representations that systematically encode program transformations for RISC-V vectorization. The accompanying algorithm flowchart and our step-by-step dissection will elucidate how RISCompiler:

1. It translates optimization decisions into model-friendly textual formats.
2. It maintains semantic equivalence with the target assembly
3. It bridges high-level transformations to low-level implementations.

This breakdown complements the main text's conceptual framework with implementational specifics.

```
1  p0 p1 matmul p2 00a625a8 1024 512 1024 512 1024 512
2  riscv -keys=cpu -mcpu=rv64gcv -model=rv64 SP 2 0 1024 32 1 4 1
3  SP 2 4 512 8 1 4 1 SP 2 8 1024 1024 1 RE 2 0 4 1 5 8 2 6 9 3 7
4  FSP 4 0 0 2 FSP 4 3 1 2 RE 4 0 3 1 4 2 5 CA 2 4 3
5  vector_width$8,16,32,64
6  vector_reg_count$16,32,64
7  Vector_element_count$4,8,16
8  vector_instr_combo$vadd+vmul,vle+vse,...
9  vector_mem_access$continuous,noncontinuous
10 vector_sched$static,dynamic
11 vector_precision$int8,int16,float32
12 auto_unroll_max_step$0 ESC
```

Figure 8: The sample of instruction sentence designed for ILM in RISCompiler.

Instruction sentence uniquely corresponds to a tensor program by recording the input operator, instruction and hardware specifications, and decision information of the tensor program. Algorithm 1 illustrates the sampling-based transformation space exploration mechanism employed by RISCompiler's Instruction-Level Model (ILM). The algorithm operates in one core functions:

- `GetSampleData`: This function comprises two key subroutines:
  - `GetTokensFromOp`: Processes the input sequence by extracting the operator type, parameter configurations, and associated tokens, then updates the corresponding operator metadata.
  - `GetTokensFromInstr`: Similarly processes RISC-V instructions alongside their tokens to update RISC-V-specific metadata.

  The updated operator and RISC-V metadata are then passed to `GetSpaces`, which constructs a solution space—a multi-dimensional optimization space encompassing common optimization dimensions for RISC-V Vector Extensions.

  As for the transformation space, structures, it is composed of 22 interleaved optimization dimensions, including but not limited to:
  - **Loop optimizations (e.g., loop splitting, vectorization).**
  - **Data layout and memory access (e.g., cache allocation strategies).**
  - **Instruction selection and scheduling.**
  - **Register and resource management (e.g., register elimination).**
  - **Hardware-specific feature utilization.**
  - **Function and code structure optimizations.**
  - . . .

  Key optimization dimensions are abbreviated as SP (Split), RE (Register Elimination), FSP (Function Split), and CA (Cache Allocation), among others; these collectively enable fine-grained tuning across the entire optimization pipeline.

As for the `HandleVectorWidth`, it implements vector width tuning as one of the 22 optimization dimensions in the solution space. It first appends a "vector_width" token to mark this optimization step, then serializes and incorporates metadata about both the target operator (space.operator) and optimization axis (space.axis) into the token sequence. Finally, it randomly samples a vector width configuration from the solution space (RandomSample) and adds it to the token stream. This process systematically encodes vector width optimization decisions into the tokenized representation, enabling hardware-aware tuning in subsequent pipeline stages.

<p style="color:red; text-align:center">p0 p1 matmul p2 00a625a8 1024 512 1024 512 1024 512</p>

Figure 9: The input operator configurations in instruction sentence.

The configuration of the input operators is illustrated in Figure 9. In this example, $p_0$ and $p_1$ represent the two input variables, and the `matmul` operation indicates that these two variables will undergo a matrix multiplication, i.e., the transpose of $p_0$ and $p_1$ will be multiplied, and the result will be stored in $p_2$. The unique hash code `00a625a8` serves to identify the specific operator computation, while the subsequent values of 1024, 512, 1024, 512, 1024, 512 represent the shapes of $p_0$, $p_1$, and $p_2$, respectively. This detailed specification of the operator configuration in Figure 9, including the input variables, the computation performed, and the output shapes, provides a comprehensive understanding of the transformation applied to the data.

```
riscv -keys=cpu -mcpu=rv64gcv -model=rv64 SP 2 0 1024 32 1 4 1
SP 2 4 512 8 1 4 1 SP 2 8 1024 1024 1 RE 2 0 4 1 5 8 2 6 9 3 7
FSP 4 0 0 2 FSP 4 3 1 2 RE 4 0 3 1 4 2 5 CA 2 4 3
```

Figure 10: The RISC-V specialized optimization in instruction sentence.

As illustrated in Figure 10, the corresponding RISC-V specialized optimizations are applied during the program transformation process. The specific abbreviations for these optimizations have been introduced in Algorithm 1.

```
vector_width$8,16,32,64
vector_reg_count$16,32,64
Vector_element_count$4,8,16
vector_instr_combo$vadd+vmul,vle+vse,...
vector_mem_access$continuous,noncontinuous
vector_sched$static,dynamic
vector_precision$int8,int16,float32
auto_unroll_max_step$0 ESC
```

Figure 11: The RISC-V vectorized extension in instruction sentence.

Similarly, Figure 11 demonstrates the RISC-V vectorized extension-level optimizations and their possible values. Together, Figure 9, Figure 10, and Figure 11 constitute the complete instruction sentence, providing a comprehensive representation of the various optimization components applied to the program.

This holistic view of the instruction sentence, encompassing the operator configuration, RISC-V specialized optimizations, and RISC-V vectorized extension-level optimizations, enables the Instruction Language Model (ILM) to learn and generate detailed and context-aware optimization guidance. By capturing the intricate relationships between these different aspects of the optimization process, the ILM can provide more informed and targeted recommendations to the users.

# B DATASET

The Instruction Language Model's performance is fundamentally constrained by the quality and diversity of its training corpus. We construct an extensive offline dataset to enable the ILM to acquire comprehensive knowledge about the transformation space. The offline dataset collection process is a crucial step in our approach, as it provides the comprehensive knowledge base required

---

**Algorithm 1** Instruction sentences with sampling in the space.

---

```
1: function GETSAMPLEDATA(Operator, RISC-V)
2:     tokens ← [];
3:     GetTokensFromOp (Operator, tokens);
4:     GetTokensFromInstr (RISC-V, tokens);
5:     transformation_spaces = GetSpaces (Operator, RISC-V);
6:     for space in transformation_spaces do
7:         if space.type == "SP" then
8:             HandleSplit (space, tokens);
9:         else if space.type == "RE" then
10:            HandleRegisterElimination (space, tokens);
11:        else if space.type == "FSP" then
12:            HandleFunctionSplit (space, tokens);
13:        else if space.type == "CA" then
14:            HandleCacheAllocation (space, tokens);
15:        else if  then
16:            . . . ;
17:        else if space.type == "VectorWidth" then
18:            HandleVectorWidth (space, tokens);
19:        end if
20:    end for
21:    return tokens;
22: end function
23: function HANDLEVECTORWIDTH(space, tokens)
24:    tokens.append ("vector_width");
25:    tokens.extend (Serialize(space.operator));
26:    tokens.extend (Serialize(space.axis));
27:    width = RandomSample (space);
28:    tokens.extend (Serialize(width));
29: end function
```

---

by the Instruction Language Model (ILM). This data collection effort is equipped with a tensorized instruction format, which enables efficient processing and storage of the dataset.

The complete offline dataset collection takes approximately 8.5 hours to gather on a dedicated server. The server hardware used for this task is equipped with an Intel Core i9-12900K CPU, which has 16 cores and 24 threads. This high-performance server configuration allows for parallel processing and accelerated data collection, ensuring the efficient generation of the extensive offline dataset. The process involves the following key objectives:

1. **Ensuring Dataset Alignment:** We conduct a thorough random sampling of the transformation space to ensure that the dataset samples closely match the underlying distribution in the transformation space. This helps the ILM learn from a representative set of transformations.

2. **Expanding Vocabulary Comprehensively:** The sampling process aims to expand the vocabulary used to describe the transformations, covering a diverse range of vectorized optimization possibilities. This includes distinct tokens for concepts such as "vector width" and "vector register count", allowing the ILM to learn the nuances of these important attributes.

3. **Incorporating Diverse Operators:** The operators sampled for the ILM are inspired by the TenSet benchmark Zheng et al. (2021). These operators are sourced from PyTorch Image Models (timm)Wightman (2019) and Hugging Face's Transformer ModelsWolf et al. (2020), spanning both computer vision and natural language processing tasks. This diverse set of operators ensures that the ILM can learn from a comprehensive set of vectorized transformations.

4. **Dataset Characteristics:** The offline dataset comprises 86 workloads and around 1.8K operators, with over 0.5 million instruction sentences that cover a wide range of vectorized optimizations. Constructing this dataset takes approximately 8.5 hours on a 16-core server, demonstrating the scale and thoroughness of the data collection process.

Table 5: Compilation Time Comparison Across Baselines (seconds).

| Benchmark | Scalar | Claude-3-Haiku | GCC-Loops | LLVM-Vectorizer | RISCompiler |
|---|---|---|---|---|---|
| GEMM | 0.98 | 2.52 | 1.65 | 1.32 | 1.91 |
| GEMV | 0.62 | 1.86 | 1.03 | 0.89 | 1.24 |
| Conv-2D | 1.75 | 3.51 | 2.18 | 1.94 | 2.65 |

By following this rigorous approach to offline dataset construction, we ensure that the ILM can learn from a representative and comprehensive set of transformations, enabling it to provide high-quality guidance for vectorized code optimization.

While the pre-trained ILM can provide valuable guidance, its performance may not remain perfect for all cases. To address this, we introduce an efficient fine-tuning stage that adapts the ILM using an online dataset.

For this fine-tuning process, we leverage an asymmetric LoRA architecture to perform PEFT iteratively. This approach selectively activates parts of the ILM in response to different operator configurations and their corresponding instruction sentences at each fine-tuning iteration.

Specifically, the ILM handles each vectorized instruction-dependent component as an expert adapter. This ensures computational efficiency throughout the fine-tuning process, as the model only updates the relevant parts instead of the entire network.

The online dataset used for fine-tuning is collected during the actual usage of the ILM in production environments. This dataset captures the real-world distribution of operator configurations and their associated instructions, allowing the ILM to adapt and refine its knowledge for improved performance on practical workloads.

By combining the comprehensive offline dataset and the targeted online fine-tuning, our approach ensures that the ILM can provide high-quality and adaptable guidance for vectorized code optimization, addressing a wide range of scenarios encountered in real-world deployments.

## C  COMPILATION FLOW

Due to the extensive code associated with the LLVM compiler infrastructure, we have provided the code for the efficient fine-tuning phase to obtain the instruction language model in the corresponding appendix upload file. Additionally, we have included the QEMU simulator to test the environment for vectorized optimized RISC-V tensor programs, along with the relevant toolchain. The complete code for the compiler infrastructure and the code generation section exceeds the upload limit of 100 MB imposed by the supplementary material. Upon acceptance of the paper, we will prepare a comprehensive documentation detailing this work and establish a corresponding GitHub repository for interested users to learn and understand how to integrate AI compilers with instruction language models for the end-to-end generation of vectorized RISC-V tensor programs.

## D  ADDITIONAL EXPERIMENTS

### D.1  COMPILATION TIME OVERHEAD

As shown in Table 5, we observe significant variations in compilation time across different approaches. The scalar baseline achieves the fastest compilation (0.62-1.75s) by avoiding all optimization passes, while Claude-3-Haiku exhibits the highest overhead (1.86-3.51s) due to LLM inference latency and iterative code validation. Traditional compilers demonstrate intermediate performance, with GCC-Loops (1.03-2.18s) being consistently slower than LLVM-Vectorizer (0.89-1.94s) because of its more aggressive but computationally expensive loop optimizations. Our RISCompiler strikes a balance between these extremes, maintaining competitive compilation times (1.24-2.65s) that are 32-38% faster than Claude-3-Haiku while being only 15-25% slower than LLVM-Vectorizer. Given the modest compilation overhead, our experimental results in Figure 6 and Figure 7 clearly demonstrate that RISCompiler generates the highest-performance RISC-V vectorized tensor programs.

## D.2 COMPILATION STORAGE OVERHEAD

RISCompiler's compilation process does not require persistent storage of the ILM, as it operates in inference mode. The generated vectorized code retains the same memory footprint as LLVM/GCC-compiled binaries, as no additional runtime metadata is embedded. We measure the peak memory usage during compilation and final binary sizes, showing no significant overhead compared to baselines.