# OpenReview forum: "Leveraging Instruction Language Model to Generate Vectorized RISC-V Tensor Programs"
_ICLR.cc/2026/Conference — Submitted to ICLR 2026_

### Official Review · Reviewer_aCLZ · 2025-10-27

**Soundness:** 3
**Presentation:** 1
**Contribution:** 3
**Rating:** 4
**Confidence:** 3

**Summary:**

The paper proposes  a way to turn tensor programs into vectorized programs using language models.  Hand-crafting vectorized programs is hard as it requires in-depth knowledge of specific instructions and is prone to error. Their contribution involves creating an instruction sentence representation suitable for language models which includes transformation details to accurately represent vectorized RISC-V tensor programs. They train an instruction language model (ILM) to generate transformations and use a code generator to translate these transformations into actual vectorized RISC-V assembly

**Strengths:**

- The paper proposes an elegant approach to generate vectorized code, avoiding the difficulty of producing thousands of tokens error-free in a single pass.
- The method achieves strong performance across multiple setups, outperforming widely used and well-known baselines.

**Weaknesses:**

- The paper is difficult to follow, as some details that would aid understanding are omitted (See questions).
- The novelty of their fine-tuning algorithm is unclear. It appears to share many similarities with *Mixture of LoRA Experts (MoLE; ICLR 2024)*, which is not cited, making it hard to assess its effectiveness relative to *MoLE*.

**Questions:**

- There are many places where \citep should be used; the current citations are inconsistent and impair readability.
- Why was *LLaMA 3.2 3B* chosen instead of a code-specific LLM of comparable or smaller size (e.g., *StarCoder-2 3B*), which may be more suitable for the task? How might the choice of model (size, training data, etc.) impact results?
- The *ILM* is fine-tuned to produce instruction sentences, right? Could you clarify what the input consists of, $r^{(i)}$, its specifications, or both?"
- Line 348: Could you clarify this statement? Instead of generating something like *I=1024|width|i.0=32|register_count|i.1=2|element_count|i.2=4|J=1024|width|j.0=32| ...</ s>* in one shot, is it generated step by step for *i*, *j*, and *k*?
- Line 371: Could you clarify what you mean by *The baseline methodology incorporates the Claude-3-Haiku with Poe without any fine-tuning technique as its central component to generate programs with vectorized programs.*? Are you simply prompting the model in a zero-shot fashion to obtain vectorized programs?
- Isn't *Claude* or any other API-available LLM compatible with your framework? Do you think it would be possible, using prompting, to leverage an instruction-tuned LLM as both *ILM* and *code generator* to achieve better performance?
- Line 390: Could you provide a citation for GPT-2?
- Figure 6: Do you have any intuition why **DEP** seems more difficult to speed up?
- How critical is LoRA to your framework?

---

### Official Review · Reviewer_vVEt · 2025-10-31

**Soundness:** 2
**Presentation:** 2
**Contribution:** 2
**Rating:** 2
**Confidence:** 3

**Summary:**

The paper introduces RISCompiler, which turns RVV auto vectorization into an LM generation task using compact instruction sentences and an asymmetric fine tuning scheme. It reports speedups over GCC/LLVM autovectorizer and a general LLM baseline on single operators and DNNs, using QEMU for validation.

**Strengths:**

•	Clear pipeline and interesting contribution.
•	Broad benchmarks across ops and DNNs.

**Weaknesses:**

1. All performance results are validated and timed under QEMU with RVV. While this comparison provides early evidence that the method is good, QEMU is not accurate and does not model many microarchitectural effects. Thus, the reported speedups may not translate to real hardware. Adding evaluations on real hardware would substantively strengthen claims.
2. Comparisons to GCC/LLVM autovectorizer are necessary but not sufficient. The closest baselines include LM-driven tensor program generators. The most important being TLM/OSDI’24 (“Enabling tensor language model to assist in generating high-performance tensor programs for deep learning”). Why aren't you comparing to TLM?
3. How does the proposed method guarantee correctness? The text says: “Should the transformation be deemed invalid, the ILM discards the programs and commences the regeneration”. How is that performed? This is ambiguous in the paper.

**Questions:**

•	How do you guarantee the correctness of vectorized outputs?
•	Why didn’t you evaluate on real hardware?
•	Why not compare with TLM?

---

### Official Review · Reviewer_ZoQu · 2025-10-31

**Soundness:** 2
**Presentation:** 2
**Contribution:** 2
**Rating:** 4
**Confidence:** 4

**Summary:**

The paper proposes RISCompiler, a compiler that uses a language model (an “Instruction Language Model” or ILM) to automatically generate vectorized RISC-V tensor programs. The end goal is to make RISC-V programs faster by automatic vectorization using the RISC-V vector extension (RVV). Since manual vectorization is complex, traditional compilers oftenm miss potential optimizations. ILM represents RISC-V programs as instruction setences (short, LM-friendly text descritpions of code transforms). The authors train the model with a large offline dataset of vectorized program samples and let it generate optimized vectorized code.
After LORA-finetuning Llama 3.2-3B, RISCompier obtains significant;y faster programs.

**Strengths:**

- Novelty: First attempt (to my knowledge) to treat auto-vectorization as a text generation problem using a language model. However, note later on the weaknesses: there has been important work on LLMs for compilers and compiler optimization that this work is missing.
- Practical relevance: Targets the RISC-V vector ISA, which is increasingly important for open hardware.
- Interesting, novel approach based on a 3 stage pipeline. Smart decomposition rather than end-to-end.
- Efficient deep learning: small model and uses LoRA.
- The speed numbers are non-trivial.

**Weaknesses:**

- It's not clear to me that the baselines are the real baselines you'd have in a real-world scenario. The results show that this method can indeed auto-vectorize scalar tensor programs, and this naturally leads to speedups, but there's a lack of evaluation on more alternatives to vectorize.
- Shallow description of the dataset.
- This paper misses related work on language models for compilation and compilation optimizations, some of these works are critically related for understanding the novelty: https://arxiv.org/abs/2407.02524, https://arxiv.org/abs/2309.07062, https://arxiv.org/abs/2108.07639, https://arxiv.org/abs/2309.14396, https://openreview.net/forum?id=LWfDcI6txJ, https://arxiv.org/html/2412.12163v1, https://arxiv.org/abs/2410.08806. Cummins et al trained an LLM to directly perform compiler optimizations.
- I'm not a big fan of introducing new terminology such as Instruction Language Model. Couldn't you use terms already used in related work?
- The evaluation is limited, only a few benchmarks.
- The method section is overly dense and not written very clearly.

Misc:
- The title misses either an article or ILM should be plural.
- “which are capable” -> “which is capable.”
- Inconsistent capitalization: “Instruction language model” vs “instruction language model.”
- Missing articles: e.g., “design our generation-based RISC-V auto vectorization compiler utilizing instruction language model” -> "design a generation-based RISC-V auto-vectorization compiler utilizing an instruction language model.”

**Questions:**

Will you open-source the code, checkpoints, and/or dataset?

How much work do you estimate this would take to adapt to a different ISA?

Did you consider RL rather than supervised learning?

---

### Official Review · Reviewer_cjDV · 2025-11-01

**Soundness:** 1
**Presentation:** 1
**Contribution:** 2
**Rating:** 0
**Confidence:** 3

**Summary:**

The paper proposes a solution to find the optimal vectorization configuration for each loop in a program to obtain optimal performance on a RISC-V processor.
The solution involves representing the program and/or vectorization and compiler configuration in a certain textual format, followed by training or finetuning a model.
The results show a speedup of 1.6x compared to existing RISCV compiler.

**Strengths:**

Results show 1.6x speedup compared to existing C++ compilers.
Performance evaluated on end-to-end neural network models.
Rather than just feeding assembly or C++ programs directly into an LLM, authors try to find efficient textual representation.

**Weaknesses:**

The paper is not clear, and has many gaps.
It is not clear to me whether the approach trained a model from scratch or finetuned.
MoEs are out of the blue mentioned in the "Code Generation Stage" but not mentioned in the subsequent training stages. It is not clear whether the original model had an MoE structure. Then in the subsequent sections that were supposed to explain the solution in detail, nothing was mentioned about a MoE architecture or how each expert was trained.
The evaluation section did not compare with ML-based approaches for compiler optimization. I understand that papers on ML for compilers tend to not follow a standard dataset, so it is difficult to find a paper that evaluates on the same task, hardware backend, program types, etc. to compare against. However, the authors could have trained their dataset using an architecture or approach proposed on another ML for compilers paper, or could have evaluated brute force auto-tuning approaches, or auto-tuning that exists in a compiler like TVM.

**Questions:**

- Does the solution train a model from scratch or start from a pretrained model? What is the architecture of the model? What were the training hyperparameters?
- Figure 3 is not clear to me. Figure 5 that shows how data is fed into the LLM, does not show the usage of data presented in Figure 3.
- Are the representations extracted from an LLVM IR representation of the code samples?
- I suggest comparing with Meta's LLM Compiler ( https://arxiv.org/abs/2407.02524 ). Although it was trained for a different task (minimize IR instruction count), the model is open source and could be evaluated to see what speedup (or slowdown) it leads to on the evaluate data
- I also suggest to compare using Tiramisu ( https://tiramisu-compiler.org/ ). I believe it can evaluate similar program samples that this paper evaluated on
- The motivation behind the representation of data was not clear to me, and whether there is any novelty or significance to warrant publication in a top-tier ML conference is also not clear. For example, the representation of programs suggested in  https://arxiv.org/abs/2104.04955 has been backed by intuition and explanation, while this paper does not back its proposed representation or learning algorithm with a concrete explanation.

Notes:
- Figure 7: Lower than 1 performance sounds like a slowdown. Maybe the y-axis should be labeled "Normalized Latency"
- Line 431: "mincycle": need to fix formatting

---

### Meta-Review · Area_Chair_txPp · 2026-01-07

**Summary:**

The paper proposes RISCompiler, a framework that leverages Instruction Language Models (ILMs) to generate vectorized tensor programs for RISC-V. While the reviewers acknowledged the novelty of treating vectorization as a text generation task and the clean design of the pipeline, the consensus is to Reject. This decision is primarily informed by three critical concerns shared across the reviews:

Methodological Flaw in Evaluation: The performance evaluation relies exclusively on QEMU simulation. As noted by Reviewer vVEt, QEMU cannot accurately model microarchitectural effects, meaning the reported speedups are unreliable and may not translate to real hardware.

Missing Critical Baselines: The paper fails to compare against relevant state-of-the-art ML-based compiler frameworks (such as TLM and Meta's LLM Compiler), comparing only against standard GCC/LLVM. This makes it impossible to assess the true contribution relative to the field.

Clarity and Technical Inconsistencies: There are significant unresolved ambiguities regarding the technical details, specifically the inconsistency between the "MoE" architecture mentioned in the text and the training description, as well as the unclear correctness guarantees for the generated code.

**Reviewer Concerns:**

Despite the interesting premise of applying Large Language Models (LLMs) to compiler optimization, the reviewers were unanimous in their criticism, pointing out several fundamental flaws in the evaluation and presentation.

Outstanding Concerns:

Insufficient Baselines: A critical recurring theme across reviews is the lack of comparison against state-of-the-art ML-based compiler frameworks. Reviewer vVEt highlighted the absence of TLM (Tensor Language Model), a closely related baseline. Similarly, Reviewer cjDV noted the omission of comparisons to Meta’s LLM Compiler and Tiramisu. Reviewer ZoQu also pointed out missing literature on LLMs for code optimization. Comparing only against standard GCC/LLVM auto-vectorizers was deemed insufficient for a contribution in this specific domain.

Evaluation Methodology (QEMU vs. Hardware): Reviewer vVEt raised a fatal flaw regarding the performance metrics: all results were validated and timed using QEMU rather than real hardware. As QEMU does not accurately model microarchitectural effects, the reported speedups are unreliable and may not translate to physical RISC-V silicon.

Clarity and Technical Inconsistencies: The paper suffers from significant clarity issues. Reviewer cjDV discovered a discrepancy where "MoEs" (Mixture of Experts) were mentioned in the text but completely absent from the training description. Reviewer aCLZ found the novelty of the fine-tuning approach unclear, noting unaddressed similarities to "Mixture of LoRA Experts" (MoLE).

Correctness Guarantees: Reviewer vVEt questioned how the method guarantees the correctness of the vectorized outputs, noting that the explanation of how invalid transformations are discarded is ambiguous.

**Reviewer Scores:**

Reviewer cjDV (Score: 0): Would likely remain at a Strong Reject due to fundamental issues with clarity ("poor presentation") and the lack of standard ML-compiler baselines.

Reviewer ZoQu (Score: 4): Might have considered a higher score if the related work and baselines were addressed, but currently views the evaluation as limited and the method description as "overly dense".

Reviewer vVEt (Score: 2): Unlikely to change score without new experiments on real hardware, which is the reviewer's primary critique.

Reviewer aCLZ (Score: 4): Acknowledged the "elegant approach" but remains hesitant due to the unclear novelty of the fine-tuning algorithm vs. MoLE and poor presentation.

---

### Decision · Program_Chairs · 2026-01-26

Reject